# Bioluminescence Imaging and ICP-MS Associated with SPION as a Tool for Hematopoietic Stem and Progenitor Cells Homing and Engraftment Evaluation

**DOI:** 10.3390/pharmaceutics15030828

**Published:** 2023-03-03

**Authors:** Murilo M. Garrigós, Fernando A. Oliveira, Mariana P. Nucci, Javier B. Mamani, Olívia F. M. Dias, Gabriel N. A. Rego, Mara S. Junqueira, Cícero J. S. Costa, Lucas R. R. Silva, Arielly H. Alves, Nicole M. E. Valle, Luciana Marti, Lionel F. Gamarra

**Affiliations:** 1Hospital Israelita Albert Einstein, São Paulo 05652-000, SP, Brazil; 2LIM44—Hospital das Clínicas da Faculdade Medicina da Universidade de São Paulo, São Paulo 05403-000, SP, Brazil; 3Center for Translational Research in Oncology, Cancer Institute of the State of Sao Paulo—ICESP, São Paulo 01246-000, SP, Brazil

**Keywords:** hematopoietic stem cells, bone marrow transplantation, bioluminescence imaging, ICP-MS, superparamagnetic iron oxide nanoparticles, engraftment, homing

## Abstract

Bone marrow transplantation is a treatment for a variety of hematological and non-hematological diseases. For the transplant success, it is mandatory to have a thriving engraftment of transplanted cells, which directly depends on their homing. The present study proposes an alternative method to evaluate the homing and engraftment of hematopoietic stem cells using bioluminescence imaging and inductively coupled plasma mass spectrometry (ICP-MS) associated with superparamagnetic iron oxide nanoparticles. We have identified an enriched population of hematopoietic stem cells in the bone marrow following the administration of Fluorouracil (5-FU). Lately, the cell labeling with nanoparticles displayed the greatest internalization status when treated with 30 µg Fe/mL. The quantification by ICP-MS evaluate the stem cells homing by identifying 3.95 ± 0.37 µg Fe/mL in the control and 6.61 ± 0.84 µg Fe/mL in the bone marrow of transplanted animals. In addition, 2.14 ± 0.66 mg Fe/g in the spleen of the control group and 2.17 ± 0.59 mg Fe/g in the spleen of the experimental group was also measured. Moreover, the bioluminescence imaging provided the follow up on the hematopoietic stem cells behavior by monitoring their distribution by the bioluminescence signal. Lastly, the blood count enabled the monitoring of animal hematopoietic reconstitution and ensured the transplantation effectiveness.

## 1. Introduction

The hematopoietic stem cells (HSCs) are multipotent primitive cells that can self-renew and differentiate into several hematopoietic lineages, in addition, less undifferentiated cells as hematopoietic progenitor cells are also able to proliferate and differentiate in several differentiated lineage cells (HPCs) [1,2]. Hematopoietic stem and progenitor cells (HSPCs) are mainly allocated in a microenvironment called hematopoietic niche found in the bone marrow (BM); however, they may also be found in the peripheral blood (PB) and umbilical cord blood [3]. Because of these cells’ ability to repopulate a lethally irradiated bone marrow, it is possible to transplant them in cases where there is partial or complete compatibility between donor and recipient, in a procedure known as bone marrow transplantation (BMT). The BMT is one of the many treatments for various hematological and some non-hematological disorders [4,5].

Once transplanted, the HSC are compelled to migrate to the hematopoietic niches present in the BM by a process known as homing [6]. The homing is a rapid process that occurs through the interaction between chemokines and their receptors starting an intracellular signaling that leads to a cell directional migration [7]. One of the main interactions involved in the HSCs homing includes the axis CXCL12:CXCR4. CXCL12 is also known as chemokine stromal cell-derived factor-1 (SDF-1) which is produced by the stromal cells that constitute the hematopoietic niches which is the ligand of C-X-C chemokine receptor type 4 (CXCR4) present in the HSC [8,9,10]. This interaction plays a key role in guiding and retaining HSC in the BM after BMT [7,8].

Some studies sought to better understand the dynamics of migration and engraftment of HSC in vivo using molecular imaging techniques [11]. One viable alternative for increasing the knowledge about the HSC migration process in a quantitative way is the use of nanoparticles. Nanoparticles have been developed with various structures for biomedical applications. However, in order to be used as tracking agents, these particles must present some essential properties, such as adequate polydispersion, high stability, and biocompatibility [12,13]. Among the most used nanoparticles, one that stands out is the superparamagnetic iron oxide nanoparticles (SPION), which may act as contrast in magnetic resonance imaging (MRI) [14]. These nanomaterials are a unique tool that assists the understanding of cell migration due to their superparamagnetic properties and reliable traceability [13]. 

SPION may also be functionalized with fluorescent or radioactive agents, and operates as MRI contrast [15,16,17]. Another alternative for tracking cells labeled with SPION is quantifying iron from SPION internalized in cells and present in tissues after transplantation by inductively coupled plasma mass spectrometry (ICP-MS) [18]. This spectrometric technique can detect elements with high sensitivity [19]. The association of ICP-MS with molecular imaging techniques such as bioluminescence imaging (BLI) may provide new insights into the migration process of HSPC into the tissues of interest after BMT.

Therefore, the present study proposes to evaluate an alternative method using HSPC labeled with SPION and their quantification by ICP-MS for HSPC homing assessment, combined with BLI for biodistribution and engraftment evaluation.

## 2. Materials and Methods

### 2.1. Ethical Statement and Animal Model

Thirty-three mice approximately eight weeks old from the Instituto de Ciências Biomédicas (ICB) from Universidade de São Paulo (São Paulo, Brazil) vivarium were allocated into Centro de Experimentação e Treinamento em Cirurgia (CETEC) from Hospital Israelita Albert Einstein (São Paulo, Brazil), a vivarium accredited by the Association for the Assessment and Accreditation of Laboratory Animal Care International (AAALAC International) and maintained at 21 ± 2 °C and 60% ± 5% relative humidity with full ventilation, under a 12 h light/dark cycle (7 a.m.–7 p.m.), and they had access to food and water ad libitum, being such conditions monitored daily. The use of animals in this study was approved by the Ethics in Animal Research Committee of the Hospital Israelita Albert Einstein, under the number 4932/21.

### 2.2. In Vitro

#### 2.2.1. Aspiration and Isolation of HSPC from Animals Treated with 5-FU

For the aspiration and isolation of HSC, four days before the procedure, 150 mg/kg of 5-Fluorouracil (5-FU; EMD Millipore Corp., St. Louis, MO, USA) was administrated in twelve animals. These animals were euthanized by anesthesia overdose and their femurs and tibias were extracted.

All the muscle tissue was removed to expose the bone. The bone epiphyses were removed and using a 1 mL syringe, the bone cavity was rinsed with StemSpan Serum-Free Expansion Medium (SFEM; Stem Cell Technologies, Vancouver, BC, Canada) supplemented with 100 U/mL of penicillin, 100 µg/mL of streptomycin, and 0.25 µg/mL of Gibco Amphotericin B (GIBCO—Invitrogen Technologies, New York, NY, USA). The material obtained from the bone cavity was filtered in a 40 µm Cell Strainer (Corning, New York, NY, USA) and centrifuged at 500× *g* for 5 min at 21 °C. Following centrifugation, the pellet was resuspended in 4 mL of SFEM and carefully transferred over 4 mL of Ficoll-Paque^TM^ Premium (1.084 g/mL) (GE Healthcare, Uppsala, Sweden), then the tube was centrifuged at 400× *g* for 30 min at 21 °C. 

Red blood cells lysis was performed using 1 mL of Ammonium–Chloride–Potassium Buffer (ACK, GIBCO—Invitrogen Technologies, New York, NY, USA) into the filtered bone marrow material on ice bath for 5 min, with homogenization at every minute.

Therefore, new centrifugation was performed at 500× *g* for 5 min at 21 °C and the supernatant was discarded, followed by a new addition of ACK buffer. The pellet was resuspended in 10 mL of SFEM supplemented with interleukin 3 (IL-3) (10 ng/mL), interleukin 6 (IL-6) (10 ng/mL), and stem cell factor (SCF) (100 ng/mL) (PeproTech, Rocky Hill, CT, USA) and incubated at 37 °C, 60% relative humidity, and 5% CO_2_ throughout all the culture assays.

#### 2.2.2. Immunophenotypic Characterization of Isolated Cells from Animals Treated with 5-FU

The immunophenotypic characterization of isolated hematopoietic stem and progenitors cells (HSPC) from animals previously treated with 5-FU was performed by flow cytometry (FACSAria™ III, BD Biosciences, San Jose, CA, USA) using the BD Mouse Hematopoietic Stem and Progenitor Cell Isolation Kit (BD Biosciences, San Jose, CA, USA), with comprises the following antibodies: Sca-1-PE-Cy7, c-Kit-PE, CD34-FITC, and the Lineage-APC antibody cocktail (CD3, CD45R, Ly6C, Ly6G, CD11b and TER-119).

Isolated HSPCs were stained by incubation with the antibodies specific for HSPCs, hematopoietic lineages, and a viability dye: c-Kit, Sca-1, Lineage and the viability dye 7-Amino-Actinomycin D (7-AAD). The acquisition of at least 10,000 events were performed into a flow cytometry (FACSAria™ III, BD Biosciences, San Jose, CA, USA). The results were analyzed using FACSDIVA software version 9.0 (BD Biosciences, San Jose, CA, USA) and FlowJo version 10.6 (BD Biosciences, San Jose, CA, USA). The analyses used as a gate strategy, a first selection for singlets (FSC-H vs. FSC-A), followed by a second gate for viability (7-AAD unstained cells). Next, only viable cells were gated for granularity vs. lineage markers, and only lineage negative cells were analyzed for HSPC specific markers.

#### 2.2.3. Effects of 5-FU in BM Cellularity

Four days after 5-FU administration, animals were euthanized by anesthesia overdose, tibia and femur samples were extracted, and blood samples were smeared. The tissue samples were fixed in 4% Paraformaldehyde (Sigma-Aldrich, St. Louis, MO, USA) solution for 24 h, and the bones were decalcified in the formic acid solution for 2–8 h. In sequence, the tissues were dehydrated in absolute alcohol for 5 h (1 h for each vat), then diaphonized in three xylene batches for 3 h and immersed in paraffin for 2 h at 60 °C. Finally, the samples were sliced to a thickness of 5 µm using a Leica RM2245 microtome (Leica, Buffalo Grove, IL, USA), to perform the Hematoxylin and Eosin (H&E) staining and the images were analyzed using a Nikon TiE fluorescence microscope (Nikon, Tokyo, Japan).

#### 2.2.4. Transduction with Lentiviral Vectors for Luciferase Expression in HSPC

The HSPC transduction was performed by viruses carrying the lentiviral vector pMSCV-Luc2-T2A-Puro codifying a codon-optimized version of luciferase (kindly provided by Dr. Deivid de Carvalho Rodrigues). The production of virions used in lentiviral transduction was performed as previously described [20,21,22] and the bioluminescence mechanisms are well-established in the literature as is described in the study of Mezzanotte L. et al. [23]. After the HSPC isolation, these cells were cultured in supplemented SFEM with cytokines for 24 h after isolation and then counted in a Neubauer chamber with Trypan blue stain (GIBCO—Invitrogen Technologies, New York, NJ, USA). Next, the lentivirus particles (5 viral particles/cell) and polybrene (Sigma-Aldrich, St. Louis, MO, USA) at 8 µg/mL were added to the culture for HSPC transfection. The resulting suspension was centrifuged at 1000× *g* for 90 min at 33 °C and the pellet was resuspended in 3 mL of supplemented SFEM with cytokines, transferred to a 24-well plate, and incubated for 24 h at 37 °C, 60% relative humidity, and 5% CO_2_, allowing these cells to be tracked by bioluminescence image when associated to the D-Luciferin presence.

#### 2.2.5. Labeling of HSPC with SPION

The labeling process was performed, firstly adding the HSPC into wells with a commercial colloidal solution of SPION coated with aminosilane dispersed in an aqueous medium (Chemicell GmbH, Berlin, Germany) at an initial concentration of 25 mg/mL, a density of 1.25 g/cm^3^, a hydrodynamic diameter of 100 nm, and a number of 1.8 × 10^5^ nanoparticles/g. The HSPCs labeled with SPION were then centrifuged at 500× *g* for 5 min at 21 °C and resuspended in Phosphate Buffer Saline (PBS; GIBCO—Invitrogen Technologies, New York, NY, USA), dripped in glass slides, and left overnight. After overnight incubation, the cells were stained with drips of a solution containing 5% potassium ferrocyanide (Sigma Aldrich, St Louis, MO, USA) and 5% hydrochloric acid (Merck, Darmstadt, Germany) for 10 min and washed once with Milli-Q^®^ water (EMD Millipore Corporation, Bedford, MA, USA). This Prussian Blue staining intended to highlight the SPION internalized into HSPC. Finally, the stained slides were then observed using brightfield microscopy. 

### 2.3. In Vivo

#### 2.3.1. Bone Marrow Transplantation Model

To perform the BMT assay, twelve mice were used as hosts and irradiated 24 h before transplantation with 9 Gy in a single fraction in an X-ray radiator Rad Source RS2000 Biological System (Buford, GA, USA) of the Instituto do Câncer do Estado de São Paulo (ICESP) (São Paulo, Brazil). Therefore, a suspension of 150 µL of SFEM with 3 × 10^6^ HSPC was injected via retro-orbital plexus in anesthetized host mice using a 27 G needle at an infusion rate of 75 µL/min controlled by a micro drop infusion pump (Pump 11 Elite Nanomite, Holliston, MA, USA).

#### 2.3.2. Evaluation of Homing of HSPC Labeled with SPION by ICP-MS

##### Tissue Digestion

For homing evaluation, 24 h after the HSC transplantation, animals were euthanized by anesthesia overdose, and spleen, tibia, and femur samples were collected. The spleens samples were stored in a 10% Paraformaldehyde solution at 4 °C. The muscle tissue was removed from the tibia and femur samples. The bone epiphyses were removed and using a 1 mL syringe the bone cavity was rinsed using Milli-Q^®^ water. The material collected from BM was filtered in a 40 µm Cell Strainer.

Before the tissue digestion, the spleen samples were dried overnight at 80 °C and the dried tissue samples were digested in the Titan-MPS (Perkin Elmer, Waltham, MA, USA). The tissues were placed in vessels with a solution of 7 mL concentrated nitric acid (Sigma-Aldrich, St. Louis, MO, USA) and 3 mL 30% hydrogen peroxide (Merck, Darmstadt, Germany). Meanwhile, approximately 0.9 mL of BM material was placed in a vessel with 6.5 mL of concentrated nitric acid, and the following ramp parameters were used for digestion: 10 min to reach 170 °C and 30 bar, held for 10 min, then in 1 min up to 200 °C and 40 bar, held for more 30 min, decrease to 50 °C and atmospheric pressure in 1 min and hold for 10 min to complete the digestion.

##### Iron Quantification by ICP-MS

The quantification of the iron content of digested tissue was performed from control animals for basal iron content reference, and from animals after BMT with HSPCs labeled with SPION, after 24 h of the transplantation of SPION-labeled HSPCs, their migration to spleen and BM was evaluated using ICP-MS model NexIon 350X (Perkin Elmer, Waltham, MA, USA). Digested spleen samples were diluted 100 times, while BM samples were diluted 10 times with a 1% nitric acid solution, and the measurement of iron content in all samples were performed in sextuplicate. The calibration curve was prepared using a certified external standard for iron (NexION #N8145054). 

#### 2.3.3. Evaluation of HSPC Engraftment by BLI

The engraftment was evaluated through BLI, which was obtained using the In Vivo Imaging System (IVIS) Lumina III (Perkin Elmer, Santa Clara, CA, USA) and quantified by IVIS imaging software version 4.7.3 (Perkin Elmer, Santa Clara, CA, USA). Until the 40th day after transplantation, the images were captured weekly (4, 7, 19, 26, 33, and 40th day) and, after that, every three weeks (61, 82, 103, and 124th day). Right after the BMT, the animals were trichotomized to reduce the absorption of radiation by the fur melanin to mitigate the signal attenuation. For image acquisition, ten minutes prior to in vivo image, D-Luciferin 150 mg/kg i.p. (XenoLight, Perkin Elmer, Boston, MA, USA) was administered in host animals, this procedure was repeated each time point of evaluation described above. The images were obtained in dorsal, ventral, and lateral positions, using the following acquisition parameters: automatic exposure time, F/stop 4, binning 8, and FOV 12.9 cm.

#### 2.3.4. Hematopoiesis Reconstitution

Hematopoiesis reconstitution was evaluated by blood count of PB samples extracted from the animal’s retroorbital plexus, using EDTA tubes. The analysis was made by a Hematoclin 2.8 vet (Bioclin, Belo Horizonte, Brazil) and was quantified: granulocytes, lymphocytes, monocytes, white blood cells (WBCs), red blood cells (RBCs), and platelets (PLTs). The samples were collected 14, 33, 61, 82, 103, and 124 days after the BMT model introduction.

### 2.4. Statistical Analysis

For each analysis, data were described as the mean and the standard deviation for at least two independent analyses with 6 mice per group. Before comparisons, the normality Shapiro–Wilk test was performed to selected the parametric or non-parametric test to compare two independent samples through the JASP software v0.11.1.0 (http://www.jasp-stats.org; accessed on 4 January 2023). The statistical significance level of the data was established as *p* ≤ 0.05 (* *p* ≤ 0.05. ** *p* ≤ 0.01, *** *p* ≤ 0.001).

## 3. Results

### 3.1. Immunophenotypic Characterization of Isolated Cells from Animals Treated with 5-FU

The immunophenotypic characterization of HSPC was performed by flow cytometry for control animals and animals treated with 5-FU administered 4 days before the sample aspiration (Figure 1). The overall analysis strategy was performed by a selection in FSC-H versus FSC-A for doublets exclusion (Figure 1A,E), followed by the selection of viable cells as shown in Figure 1B,F, excluding the 7AAD stained cells. The viable cells were then analyzed for Lineage markers, and only the cells that were negative for these markers were analyzed for undifferentiated populations. Among the undifferentiated cells, we evaluated the percentage of LK (Lineage^−^, Sca-1^−^ and c-Kit^+^ cells) and the LSK (Lineage^−^, c-Kit^+^ and Sca-1^+^ cells).

Figure 1C,G displays the results of these analyses. There is an increase in the percentage of undifferentiated cells population in BM when comparing the untreated group to those treated with the 5-FU. The control animals (untreated) Lineage negative cells constituted 12.0% of BM cell population (Figure 1C), while in the animals treated with a single dose of 5-FU (after 4 days), this population increased up to 29.8% (Figure 1G). 

Moreover, the LSK cells quadrupled with the use of 5-FU, being only 8.12% of the BM cell population in the untreated animals (Figure 1D); meanwhile, in animals treated with 5-FU, the LSK lineage population represented 34.4% (Figure 1H). This result indicates an elevation in HSC percentage after the 5-FU administration, while the percentage of mature hematopoietic cells decreased.

### 3.2. Effects of 5-FU on BM Cellularity

The histological analysis of the BM from animals of both groups, treated and untreated with 5-FU, was performed to evaluate the effects of the drug on BM cellularity. The H&E staining (Figure 2) evidenced the increase of lineage precursors cells in animals treated with 5-FU. The PB smear was also collected from both animals to evaluate possible alterations in blood cellularity.

Figure 2A exhibits the BM of a control animal with the predominance of acidophilic tissue. Higher magnifications of the same region evidence the bone trabeculae and white bloodline precursor elements (Figure 2B), in addition to bone trabeculae strewn with adipocytes, and hematopoietic precursors cell (Figure 2C). Figure 2D highlight the BM cellularity with predominance of erythrocytes, hematopoietic precursor cells, and megakaryocytes.

On the other hand, in the BM of animals treated with 5-FU, basophilic tissue is observed (Figure 2E) and it is possible to notice the presence of bone trabeculae and the predominance of hematopoietic precursor cells (Figure 2F,G). In addition to that, Figure 2H displays the predominance of precursor cells, which differs from the cellularity of the control animal, in which is observed a higher prevalence of erythrocytes. 

Figure 2I,J shows no apparent change in the PB lymphocytes or cellularity. 

### 3.3. Evaluation of HSPC Labeled with SPION and Their In Vivo Migration

After labeling the HSPC with SPION, the Prussian blue staining was performed to evaluate the nanoparticles internalization in the cytoplasm. This staining allowed to verify the presence of intracellular nanoparticles by highlighting the iron load of SPION through brightfield microscopy images (Figure 3).

The control using unlabeled cells (Figure 3A,B) was compared with HSPC labeled with different SPION concentrations. HSPCs labeled with 10 µg Fe/mL presented a small number of labeled cells (Figure 3C,D), HSPCs labeled with 30 µg Fe/mL displayed a higher load of iron (Figure 3E,F). However, with 50 µg Fe/mL, besides the higher number of labeled cells, there was an expressive amount of extracellular SPION (Figure 3G,H). Thus, we chose to continue the experiments using HSPCs labeled with 30 µg Fe/mL.

After 24 h of the transplantation of SPION-labeled HSPCs, their migration to spleen and BM was evaluated. Thus, tissue samples were collected, digested, and the quantification of iron was performed in both tissues and based in the calibration curve displayed in Figure 3I. Figure 3J compares the spleen iron load of control animals to the spleen of HSPC-SPION-labeled (30 µg/mL) animals. In the control animals, the iron load of the spleen was 2.14 ± 0.66 mg Fe/g, while in the group that received labeled cells, it was 2.17 ± 0.59 mg Fe/g. Therefore, no significant difference was found between these groups (*p* = 0.857), by non-parametric independent *t*-test. One of the groups showed non-normal distribution according to the Shapiro–Wilk test (*p* = 0.029).

On the other hand, the iron load in the BM of the control animals was 3.95 ± 0.37 µg Fe/mL, and in the HSPC-SPION-labeled transplanted animals it was significantly higher with values of 6.61 ± 0.84 µg Fe/mL (*p* = 0.004) by parametric independent *t*-test, as shown in Figure 3K. Both groups presented a normal distribution (*p* = 0.613 and *p* = 0.603, respectively).

Regarding the spleen, there was no significant difference between groups; therefore, it was not possible to establish iron load variation. Instead, in the BM, it was possible to estimate the non-endogenous iron, coming from the transplanted SPION-loaded HSPCs. According to Figure 3K, there were higher amounts of iron (2.66 µg Fe/mL) in BM from animals that received HSC-SPION loaded than in the BM of the control group. Based on these values, the number of nanoparticles present in BM was estimated through the following equation.
(1)$$\mathrm{SPION}\;\mathrm{number}=\frac{6\eta (A\bar{M_{x}})}{\pi \rho_{x}\phi_{x}^{3}\bar{M_{\mathrm{Fe}}}}$$
where η is the number of moles of iron, A is the atomic mass of Fe^56^ (A = 55.84 g/mol), $\bar{M_{x}}$ is the molecular weight of Fe_3_O_4_ ($\bar{M_{x}\;}=231.52\;g/\mathrm{mol}$), $\rho_{x}$ is the density of Fe_3_O_4_ ($\rho_{x}$ = 4.9 × 10^6^ g/m^3^), $\phi_{x}$ is the nanoparticles diameter ($\phi_{x}=100\mathrm{nm})$ and $\bar{M_{\mathrm{Fe}}}$ is the molecular weight of Fe_3_. The uncertainty of the number of nanoparticles present in BM was determined by:(2)$$\sigma_{\mathrm{SPION}\;}=\frac{6(A\bar{M_{x}})}{{\pi \rho }_{x}\bar{M_{\mathrm{Fe}}}}\;{\left({\left(\frac{\sigma_{\eta }}{\eta }\right)}^{2}+9{\left(\frac{\sigma_{\phi }{}_{x}}{\phi_{x}}\right)}^{2}\right)}^{1/2}$$

According to the values of Figure 3K and Equations (1) and (2), the number of SPION present in BM from transplanted animals was 1.43 ± 0.49 nanoparticles/mL (4.76 ± 1.63 moles/mL).

### 3.4. Evaluation of HSPC Engraftment by BLI

The evaluation of HSPC biodistribution and engraftment was performed by BLI through periodical quantification of bioluminescence signal intensity in in vivo images and a final ex vivo image (Figure 4). According to the BLI quantification, the maximum signal intensity was obtained around the 33rd day after transplantation and then declined, remaining constant after the 82nd day. In addition, the analysis of in vivo images showed the hematopoiesis occurrence after the BLI maximum signal intensity on the 33rd day, since the signal was no longer coming from few regions (spleen and spinal cord), but was now coming throughout the whole body of the animal.

Figure 4A displays the BLI signal evaluation of transplanted HSC engraftment over time. On the 4th day after the HSPC transplantation, the BLI signal was detectable though at a very low intensity of 5.75 ± 2.56 × 10^6^ photons/s. Three days later, on the 7th day, there was an expressive increase in the BLI signal intensity to 3.41 ± 1.65 × 10^8^ photons/s, that continued increasing to the 19th day with a value of 3.76 ± 1.89 × 10^9^ photons/s, and on the 33rd day reached the highest BLI signal intensity of 6.41 ± 3.89 × 10^9^ photons/s. After this point on the 33rd day, the BLI signal distribution was modified and the intensity reduction initiated until the 82nd day, when the signal reached a plateau of 2.35 ± 0.95 × 10^9^ photons/s with small oscillations, with a small increase on the 103rd day to a 2.01 ± 1.44 × 10^9^ photons/s. On 124th day, the signal returned to decline 1.99 ± 2.88 × 10^9^ photons/s.

Figure 4B exhibits on the left side the in vivo image of the animals in dorsal position to evaluate the biodistribution and engraftment of the HSPC after BMT, highlighting the target organs, and on the right side, the ex vivo images obtained after 124 days after the HSC transplantation from the following organs: spleen, liver, heart, lung, kidney, hind leg, spinal cord, and brain.

### 3.5. Hematopoiesis Reconstitution

Blood counts of transplanted animals were monitored over time and the fluctuations of granulocytes, lymphocytes, monocytes, RBC, PLT, and WBC were examined over a period of 124 days following BMT to evaluate the hematopoiesis reconstitution (Figure 5).

Figure 5A demonstrates a reduction in granulocytes numbers 14 days after BMT, though after 33 days, the baseline values were reestablished and remained similar until 124 days after transplantation. As for the lymphocytes (Figure 5B), 14 days after BMT there was an expressive decrease in the cell number, which was reestablished only around the 61st and 124th days. Figure 5C also displays an expressive reduction of monocyte numbers on 14th day, the baseline values after a slow recovery were restored on the 82nd day. The RBC number rapidly decreased in the first 14 days, as shown in Figure 5D, only returning to the baseline values 82 days after transplantation.

However, in Figure 5E, the number of PLTs also decreased until the 33rd day, when they reached their lowest values (91.33 ± 67.27 × 10^3^ cells/µL); after that, the PLT numbers increased and, after the 61st day, reached a plateau a little below the baseline values. Lastly, the WBC number also presented an expressive decay in numbers on the 14th day (Figure 5F); nevertheless, their values rapidly recovered and returned to the baseline 61 days after the transplantation.

## 4. Discussion

The assessment of HSC homing and engraftment provides several challenges in the BMT model, these points are very important and can impact the overall success of a transplant process [24,25]. Bearing this in mind, the current study sought to assess the effectiveness of a combination of methods, including BLI and tissue iron load quantification by ICP-MS SPION-associated as a tool to precisely assess homing and engraftment of HSC in a murine model. 

The goal of this study was to better understand the cellular phenomena related to BMT to ensure an efficient BM repopulation, such as niche restoration. However, to obtain sufficient numbers of HSC from the donor animal, it was necessary to use a stimulus such as a 5-FU single dose, four days before the HSC collection and isolation [26,27].

The use of 5-FU was shown to benefit the cellularity on the BM, since the histological analysis displayed enhanced cellularity of bone trabeculae, in comparison with the untreated control. Yet, the evaluation of HSPCs by flow cytometry displayed an approximately fourfold increase in the number of LSK cells within the Lineage-negative cell pool for animals treated with 5-FU (34.4%) compared to the untreated control (8.12%).

In addition, other studies also reported an increase in HSC number followed by a reduction of cellularity and mononuclear cells in the BM [28] after 4 days of 5-FU administration [29]. This antimetabolite drug is widely used in the treatment of solid tumors, which play a role interrupting the cell replication during S phase of cell cycle [30,31]. Thus, 5-FU affects cells with high proliferative rates, such as tumor cells, intestine cells, and hematopoietic cells [32]. Conversely, quiescent cells are not affected by the drug, resulting in the enrichment of hematopoietic progenitors in the BM [33,34]. In addition, the reduction of mature cells triggers a fast answer in the BM by increasing HSPCs aiming to return the hematopoiesis to the steady state [29,35]. 

After HSPC obtainment, aiming to evaluate their homing and biodistribution after transplantation, SPION was internalized into these cells’ cytoplasm. This process enables non-invasive cell tracking by the molecular imaging techniques such as magnetic resonance image (MRI) and fluorescence image [11,36,37]. This study evaluated the tracking of SPION-labeled HSPCs based on the quantification of iron load in tissues in the animals transplanted with SPION-labeled HSPCs and compared to control animals. Ghosh et al. [18] also reported this alternative method to evaluate the SPION biodistribution in animals by ICP-MS. 

However, before evaluating HSPC biodistribution by ICP-MS, the SPION internalization was confirmed by brightfield microscopy image after Prussian blue staining, in which the internalized nanoparticles are highlighted in the blue color, resulting from the formation of the Fe_4_[Fe(CN)_6_]_3_ complex. This method is widely used in a range of cells, from undifferentiated cells, such as stem and progenitor cells [38,39] to tumor cells [40], and also reported in HSPC [37,41,42]. 

The labeling of HSPC with 10 µg Fe/mL of SPION displayed few labeled cells and a modest load of iron internalized into the cells’ cytoplasm. In contrast, at higher concentrations, internalization was better and there were more labeled cells. However, at the highest concentration of 50 µg Fe/mL of SPION, there were many labeled cells, but an expressive amount of extracellular iron, demonstrating that at higher concentrations of SPION, the cells became overloaded and some of the nanoparticles were unable to internalize remaining free in the culture medium [43]. 

Moreover, in a previous study, Niemeyer et al. [42] demonstrated that the labeling of human HSCs with SPION did not affect the functional capacity of these cells and did not cause cytotoxicity. Therefore, the HSPC-SPION loading at the concentrations used in this study would not significantly affect the migration and engraftment of HSPC. 

Thus, we have chosen a concentration of 30 µg Fe/mL of SPION to label HSPC in this study, which was far less than the amount necessary to trigger iron overload in the recipient animal and cause toxicity [44]. 

After these cells were properly labeled, homing, biodistribution, and engraftment were evaluated in the BMT model through in vivo HSPC tracking by BLI, another non-invasive technique widely used for tracking cells in vivo [45,46,47,48]. The BLI technique was associated with the ICP-MS a technique established to quantify SPION in tissues [18,49], by quantifying and comparing the iron present in the main tissues of control animal to the animal transplanted with SPION-labeled HSPCs.

The key for BMT success is a thriving homing of the HSC to the hematopoietic sites, which influences the anchorage of cells to specialized niches and the subsequent engraftment process, which define the success rate of a transplant [25]. Shortly after the transplantation, the HSC populations tend to initially migrate to the hematopoietic organs of irradiated animals, being found mainly in the spleen and BM of these individuals [50]. This phenomenon occurs as a result of a multi-stage process involving a range of cellular signals, cytokines, cytokines receptors, and proteases, such as CXCR4, SDF-1, SCF, very late antigen 4/5 (VLA-4/5), the activation of lymphocyte function-associated antigen 1 (LFA-1), the cytoskeleton rearrangement, membrane type 1 (MT1)–matrix metalloproteinase (MMP) activation, and secretion of MMP2/9 [6,7]. 

Thus, this study evaluated the iron concentration variation in the spleen and BM of transplanted animals 24 h after HSPC infusion by ICP-MS. These sites were chosen due to their connection to the hematopoiesis process [37], as well as by the report in other studies using SPION-labeled cells and/or fluorescent agents at the same period [51].

There were no significant differences in the iron load in the spleen of controls and transplanted animal with SPION-labeled HSCs measured by ICP-MS, despite the Lanzkron et al. [50] study reporting an expressive amount of HSC homing to the spleen in a short time after transplantation. This lack of difference regarding iron in the spleen between the groups may have occurred due to the large amount of endogenous iron present in this tissue [52]. Even though the labeled cells have homing to this site, the percentual variation between the cells iron loaded and the endogenous one is extremely small and not sufficient to cause a variation between individuals, resulting in one of the limitations of using SPION for this purpose. 

On the other hand, in the BM, the amount of iron of animal controls was significantly lower (3.95 ± 0.37 µg Fe/mL) compared to animals transplanted (6.61 ± 0.84 µg Fe/mL), due to lower concentrations of endogenous iron, the ICP-MS technique proved to be sensitive. This result demonstrated that depending on the organ iron background this technique can be very promising for the tracking HSPCs in the BMT model. Moreover, the use of nanoparticles based on other metals that are less abundant in animal/human tissues may be an alternative to overcome this technique limitations. The labeling of HSPCs with gold nanoparticles, for instance, may prove to be more sensitive than the use of SPION, given the extremely low content of endogenous gold in tissues [53,54]. The change of SPION by gold nanoparticle would elicit the detection of subtle variations, allowing the detection of a small population of HSPC in any site. The use of gadolinium nanoparticles could also be another alternative, since like gold, the load of endogenous Gd is practically insignificant [55], in addition to enabling the use of MRI as a non-invasive tracking technique to corroborate the homing/biodistribution evaluation [56].

However, the tracking of SPION-labeled HSPCs only by ICP-MS impaired the assessment of the viability of anchored cells after the BMT, as iron quantification does not provide any data on cell death or proliferation [57]. Other studies have used reporter genes to overcome this limitation [58,59], and the current study used the luciferase-2 reporter gene that encodes the luminescent protein luciferase; these combined techniques allowed the evaluation of HSC homing by ICP-MS and the viability, proliferation, and engraftment by BLI. 

The BLI data showed a low-intensity signal in the BM and spleen regions four days after BMT, a signal that increased over time, reaching its peak of maximum intensity at 33 days. It was possible to observe variations in the pattern of the BLI signal over time. We have considered while the signal was limited to regions known that harbor HSC such as BM and spleen as initial biodistribution and homing. After reaching the 33rd day, the signal became distributed throughout the animal’s body, indicating recovery of hematopoiesis from the previous period. Moreover, after day 33, the signal started to decay, reaching stability at 82 days, and maintaining this pattern until the end of the evaluation period. This behavior may be explained by the fact that up to the 33rd day, the increase in the BLI signal occurred due to engraftment and proliferation of short-term HSC, which are committed to hematopoietic lineages and not being able to maintain the population of undifferentiated cells, resulting in a lower presence of transplanted cells throughout long periods. 

The stability parameter at 82 days was due to a fraction of the transplanted cell population referring to quiescent HSC, which were not replaced by the host cells and kept the signal constant, as demonstrated by Wang et al. [60], who found a similar signal pattern, with a peak of intensity in the first month, followed by a decay in the signal until reaching stability.

Additionally, the sites with highest BLI signal intensity in the BM of transplanted animals varied during analyses, as observed in the study by Cao et al. [61], which analyzed the shifting of the foci of BLI signal intensity of the HSC over time. Accordingly, in some focus the signal intensity increased, in others the signal declined, just as some foci of hematopoiesis appeared while others disappeared. This process may be a result of the non-homogeneous distribution of the hematopoiesis focus inside the BM in regeneration, varying within the BM niche and between the animals themselves [62]. 

Finally, the ex vivo BLI allowed the evaluation of the HSPC biodistribution/engraftment in each organ at the end of 124 days of evaluation. HSPCs were located mainly in the BM and spleen of the transplanted animals, corroborating the ICP-MS iron quantification results and the in vivo BLI that demonstrated the prevalence of bioluminescence signal in these sites throughout the evaluation period. 

Regarding engraftment assessment, it is notable that, even after 124 days of transplantation, a considerable part of the quiescent HSCs were located in the internal niches of the BM, as observed by the BLI signal in the spinal cord, tibia, and femur, which display an accentuated bioluminescence signal in the regions where the BM was more exposed, while in the paws, tail, and cartilaginous extremities, the signal was much lower. In addition, a small signal was found in the lungs, probably as a consequence of intravenous administration, which resulted in the retention of some cells in this region [63]. 

The blood cell analyses showed that in 14 days, all hematological parameters were below the baseline values found before the transplantation procedures. After 33 days, there was an increase in all parameters, except in platelets that returned to normality only on the 61st day of analysis. These results suggest the successful engraftment of the HSC, however, the BLI signal intensity curve that reached the maximum signal on the 33rd day after transplantation did not directly correlate to the blood counts. This may have occurred due to the repopulation of BM by host HSCs. After the total body irradiation of the mice, the donor HSCs may have acted as a support for the remaining HSCs in the hematopoietic niches of the recipient animal, which ended up competing with the transplanted HSCs, generating two cell populations in the animal, of which only one is capable of expressing BLI signal [64,65].

## 5. Conclusions

The current study established the use of ICP-MS as a viable technique for the tracking of labeled HSC with SPION and cells biodistribution, homing, and migration in the BMT model. The use of SPION directly displayed lower sensitivity due to the endogenous iron in tissues such as the spleen. Furthermore, the use of BLI as a tool for engraftment assessment displayed hgher sensitivity, and provided evaluation of hematopoiesis process over time. The association of both techniques proved to be extremely promising for the evaluation of HSC biodistribution, homing, migration, and engraftment. However, more studies are still warranted to further validate the sensitivity and future perspective of the use of ICP-MS associated with other metal nanoparticles as a tracking technique, as well as to evaluate the exocytosis of nanoparticles to circumvent possible interferences related to this technique.

## Figures and Tables

**Figure 1 pharmaceutics-15-00828-f001:**
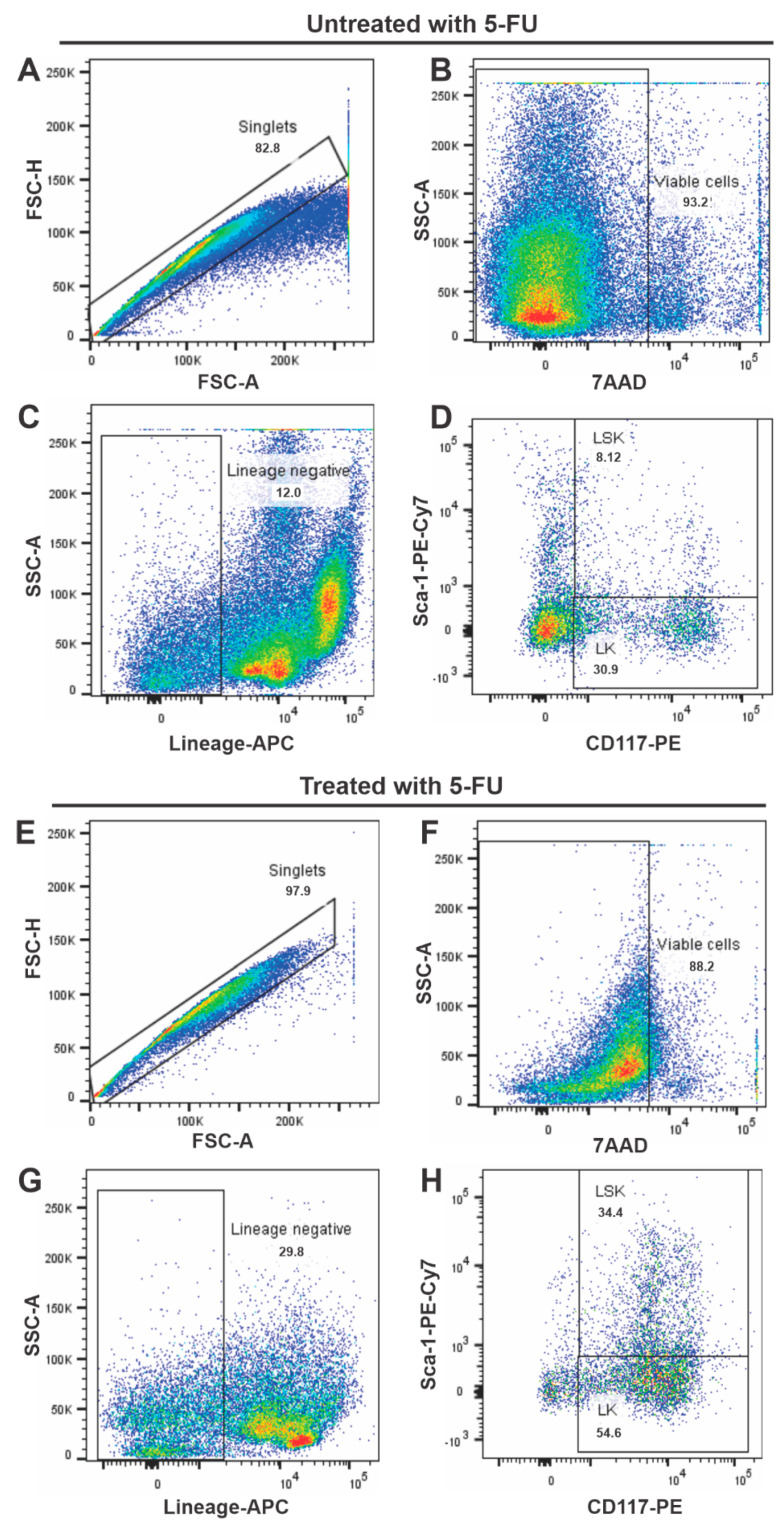
Comparative flow cytometry analysis. Analysis for singlets cells and exclusion of doublets (**A**,**E**). Analysis for viable cells (**B**,**F**). Evaluation of APC antibodies cocktail for hematopoietic cell lines (CD3+, CD45R+, Ly6C+, Ly6G+, CD11b+, and TER-119+) (**C**,**G**). LSK (Lineage^−^, c-kit^+^, and sca-1^+^), LK (Lineage^−^, sca-1^−^, c-kit^+^) of control animals (**C**,**D**) and 5-FU treated animals (**G**,**H**).

**Figure 2 pharmaceutics-15-00828-f002:**
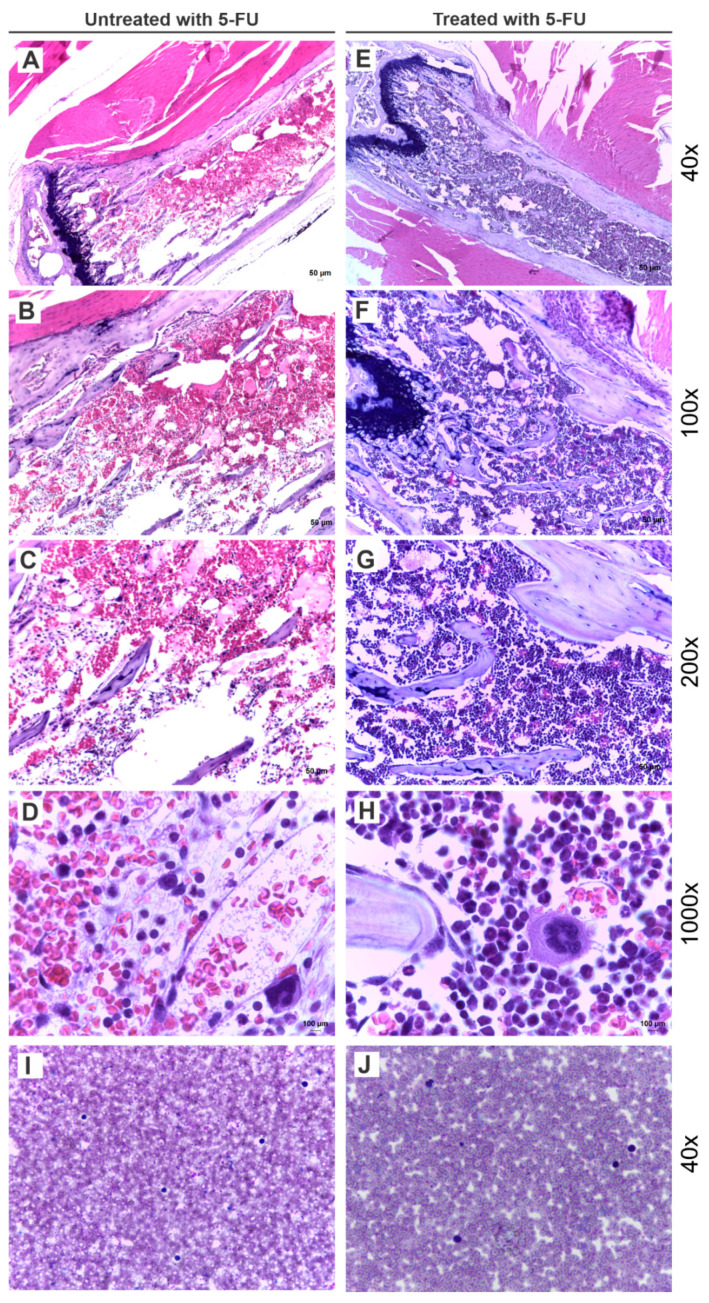
5-FU effects in bone marrow (BM) cellularity. H&E staining from BM of control animals (untreated with 5-FU) in different magnifications (40×, 100×, 200×, and 1000× respectively) (**A**–**D**). H&E staining from BM of animals treated with 5-FU 4 days after their administration in different magnifications (40×, 100×, 200×, and 1000× respectively) (**E**–**H**). Peripheral blood (PB) smear from control animal magnified 40× (**I**). PB smear from animal 4 days after 5-FU administration magnification 40× (**J**).

**Figure 3 pharmaceutics-15-00828-f003:**
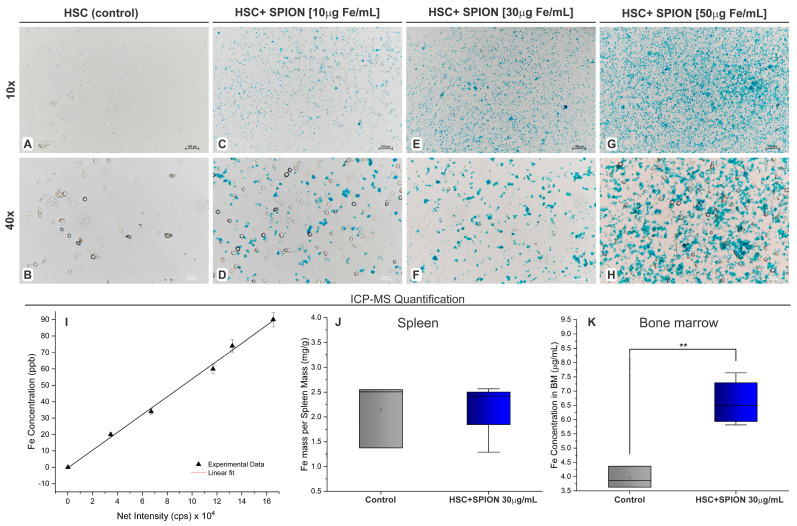
SPION internalization analysis by brightfield microscopies with Prussian blue staining and quantification of tissue iron load by ICP-MS. (**A**,**B**) Brightfield microscopies with Prussian blue staining of HSPC unlabeled in two magnifications (10× and 40×, respectively). (**C**–**H**) Brightfield microscopies with Prussian blue staining of HSPC labeled with different concentrations of SPION (10, 30, and 50 µg Fe/mL) in two magnifications (10× and 40×). (**I**) Calibration curve for iron quantification by ICP-MS for 20, 35, 60, 70, and 90 ppb of iron. (**J**,**K**) Comparison analysis of quantified iron load in the spleen and BM, respectively, of control animals and animals transplanted with HSPC labeled with 30 µg/mL of SPION. Note: ** *p* < 0.01.

**Figure 4 pharmaceutics-15-00828-f004:**
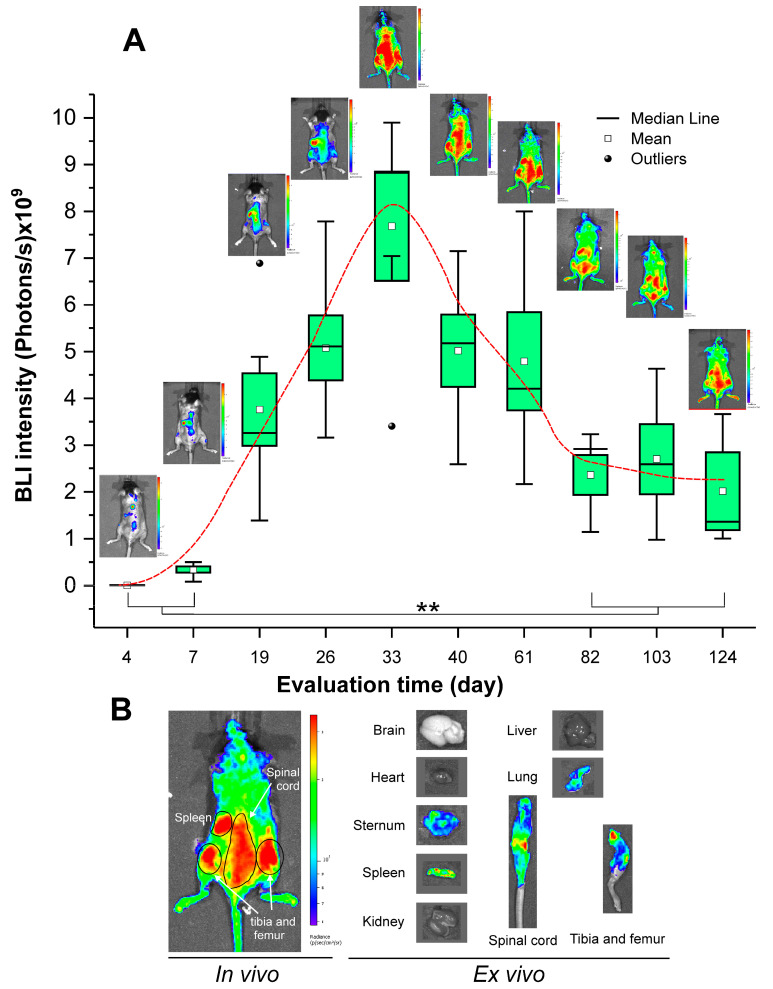
BLI assay for HSPC engraftment evaluation. (**A**) Quantification of in vivo BLI signal intensity throughout 124 days of essay and statistical comparison between the initial and final period of evaluation. (**B**) Hematopoiesis and engraftment process evaluation by the representative in vivo images and site engraftment assessment by ex vivo image. Note: ** *p* < 0.01.

**Figure 5 pharmaceutics-15-00828-f005:**
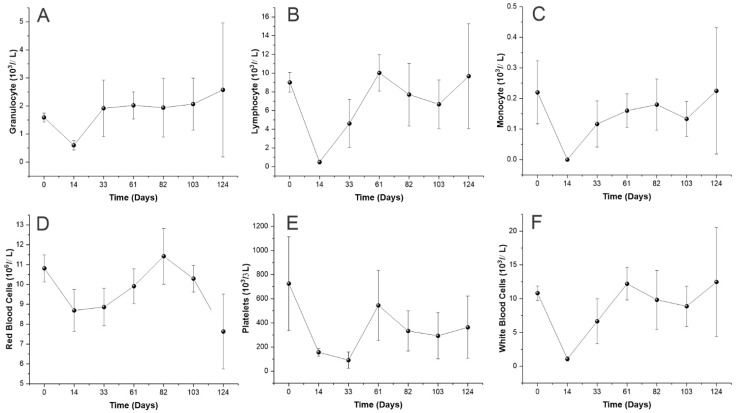
Blood count for hematopoietic reconstitution evaluation. (**A**–**F**) Variation in the number of granulocytes, lymphocytes, monocytes, red blood cells, platelets, and white blood cells, respectively, throughout 124 days after BMT.

## Data Availability

All data analyzed during this study are included in this manuscript.

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
