# Peer review of "Bioluminescence Imaging and ICP-MS Associated with SPION as a Tool for Hematopoietic Stem and Progenitor Cells Homing and Engraftment Evaluation"

_pharmaceutics, 2023, doi:10.3390/pharmaceutics15030828_

Round 1

Reviewer 1 Report

In this manuscript Garrigós and colleagues show an innovative and interesting approach to combine bioluminescence imaging and ICP-MS associated with superparamagnetic iron oxide nanoparticles (SPION) to monitor hematopoietic stem and progenitor cells (HSPC) homing and engraftment after transplantation. The assays used are quite convincing, but a deeper discussion about limitations and the applicability of this approach to different experimental systems (in vivo treatments or disease models) should be provided.

The authors need to address some issues to improve the quality of their work.

1.    The authors cannot refer to Lin- Sca1+ cKit+ (LSK) cells as HSC since this population includes also multipotent progenitors (MPP) and other less primitive progenitors. They need to change the nomenclature of HSC to hematopoietic stem and progenitor cells (HSPC) when referred to LSK. Alternatively, the analysis of LSK CD48- CD150+ Slam HSC should be provided.

2.    Does the incorporation of SPION affect HSPC function? Since the condition of iron overload was reported to change hematopoietic cell behaviour, this aspect should be investigated or at least discussed in the manuscript to avoid any misinterpretation of the results.

3.    Since the number of the analyzed mice is low (6 per group), a non-parametric statistical test would be more appropriate than t-test.

4.    The frequencies of Lineage negative and LSK in the BM of untreated mice provided in Fig. 1C-D are higher compared to what reported in recent literature. Lineage negative cells in native condition are around 1% of Tot. BM and LSK are the 0.01% of Tot. BM (1% of Lin-).

5.    The authors should check for typos and grammar errors throughout the text.

Author Response

Reviewer #1

  1. The authors cannot refer to Lin- Sca1+ cKit+ (LSK) cells as HSC since this population includes also multipotent progenitors (MPP) and other less primitive progenitors. They need to change the nomenclature of HSC to hematopoietic stem and progenitor cells (HSPC) when referred to LSK. Alternatively, the analysis of LSK CD48- CD150+ Slam HSC should be provided.

Answer:  Thank you for your suggestion, we consider the fact that LSK cells cannot be referred as HSC, therefore, when referring to LSK in the manuscript, we change it from HSC to HSPC, according to your commentary, no necessary an additional analysis suggested by the reviewer.   

  1. Does the incorporation of SPION affect HSPC function? Since the condition of iron overload was reported to change hematopoietic cell behaviour, this aspect should be investigated or at least discussed in the manuscript to avoid any misinterpretation of the results.

Answer:  We appreciate your commentary, we took into account your suggestion and added a paragraph in the discusion section of the manuscritp, about the possible affects of SPION in HSPC functio [1] and in the iron overload [2] at the eight paragraph of discussion.

  1. Niemeyer, M.; Oostendorp, R.A.J.; Kremer, M.; Hippauf, S.; Jacobs, V.R.; Baurecht, H.; Ludwig, G.; Piontek, G.; Bekker-Ruz, V.; Timmer, S.; et al. Non-invasive tracking of human haemopoietic CD34+ stem cells in vivo in immunodeficient mice by using magnetic resonance imaging. European Radiology 2010, 20, 2184-2193, doi:10.1007/s00330-010-1773-z.
  2. Atilla, E.; Toprak, S.K.; Demirer, T. Current Review of Iron Overload and Related Complications in Hematopoietic Stem Cell Transplantation. Turk J Haematol 2017, 34, 1-9, doi:10.4274/tjh.2016.0450.

  1. Since the number of the analyzed mice is low (6 per group), a non-parametric statistical test would be more appropriate than t-test.

Answer:  Thank you for your commentary, we added the information in the method section of the manuscript, about the performed a Shapiro-Wilk test to verify normality for each statistical analysis and after checking the normal distribution of all groups, we decided to the correct test for analysis, in the case highlight by the reviewer, the t-test was appropriate for analysis of the iron load in bone marrow between groups, since the both groups showed normal distribution. Furthermore, we added the value for the Shapiro-Wilk test in the result section of the manuscript.

  1. The frequencies of Lineage negative and LSK in the BM of untreated mice provided in Fig. 1C-D are higher compared to what reported in recent literature. Lineage negative cells in native condition are around 1% of Tot. BM and LSK are the 0.01% of Tot. BM (1% of Lin-).

Answer:  Thank you for your suggestion. We forgot to add one step of the isolation of the HSPC method that may be one of the causes to the higher frequency of LSK. The use of Ficoll gradient as now described in the manuscript during a second step centrifugation may have increased the relative number of LSK, resulting in a higher percentage than expect to the pool of BM cells before isolation.

  1. The authors should check for typos and grammar errors throughout the text.

Answer:  Thank you for your attention and dedication into this review, we checked for eventual typos and grammar errors throughout the text and corrected them.

Reviewer 2 Report

Comments for pharmaceutics-2235403

This manuscript proposes an alternative method for evaluating the homing and engraftment of hematopoietic stem cells using bioluminescence imaging and inductively coupled plasma mass spectrometry with superparamagnetic iron oxide nanoparticles. The study shows the effectiveness of cell labeling with nanoparticles and provides insights into the potential use of this technique for bone marrow transplantation. However, there are several issue to address:

1.      Could the authors elaborate on the method and mechanism of the bioluminescence assay they used in this work? What is the compound generating bioluminescence and how it is labeling the stem cell?

2.      Additionally, could the authors provide information on whether the luminescence signal was supposed to be stable for weeks, and explain why the signals increased and decreased over time?

3.      In addition to quantifying Fe levels in the bone marrow and spleen of transplanted animals, could the authors quantify Fe levels in different organs and tissues of the mice that received the same amount of SPION injection (without stem cell) only?

4.      How did the authors confirm that all SPIONs were inside the labeled stem cells without any remaining in the culture media? If there were SPIONs left in the media, how were they removed before transplantation?

5.      Could the authors provide a histological study of the organs and tissues from the transplanted mice to confirm the effective transplantation? This will help to further validate the effectiveness of the transplantation and the bioluminescence imaging technique used in this study.

Author Response

Reviewer #2

  1. Could the authors elaborate on the method and mechanism of the bioluminescence assay they used in this work? What is the compound generating bioluminescence and how it is labeling the stem cell?

Answer:  We appreciate your commentary and dedication into this review. We added more information about the method and mechanism of the bioluminescence assay in the method section of the manuscript according to request. The D-Luciferin substrate is commercialed by XenoLight, Perkin Elmer, Boston, MA, USA. The cell labeling process was improved in the manuscript, to turn clear the steps of process.

  1. Additionally, could the authors provide information on whether the luminescence signal was supposed to be stable for weeks, and explain why the signals increased and decreased over time?

Answer:  We appreciate your observation. We improved the method description about BLI, to turn more compreensible of temporal evaluation and the image acquisition. In addition, the BLI signal oscilation over time is depend of of the engraftment and proliferation of HSC discussed in the manuscript.

  1. In addition to quantifying Fe levels in the bone marrow and spleen of transplanted animals, could the authors quantify Fe levels in different organs and tissues of the mice that received the same amount of SPION injection (without stem cell) only?

Answer:  Thank you for your suggestion, we agree that this assay would be valid and we are even interested in carrying it out in future studies to determine the biodistribution and in vivo toxicity of SPIONs administered directly to animals. In the present study, the proposal was acute evaluate the HSC homing by ICP-MS, due to the low sensitive of BLI technique for this acute evaluation. The spleen and bone marrow organs was chose due to be the main organs with high concentration of HSC after transplantation by the literature and in our results, as shown in Figure 4B.

  1. How did the authors confirm that all SPIONs were inside the labeled stem cells without any remaining in the culture media? If there were SPIONs left in the media, how were they removed before transplantation?

Answer:  Thank you for commentary, in fact, not all SPIONs were able to internalize in the HSC, part of them remains free in solution, however, to avoid interference and consider only internalized nanoparticles, before the BMT assay, the HSC were centrifuged at 500 g for 5 minutes, as described in the methods section of the manuscript. In these centrifugation parameters, 100 nm SPION do not precipitate and remain in solution [1], therefore, when discarding the supernatant and using only the pellet formed by labeled cells, nanoparticles that did not internalize are disregarded.

  1. Dadfar, S.M.A.; Camozzi, D.; Darguzyte, M.; Roemhild, K.; Varvarà, P.; Metselaar, J.; Banala, S.; Straub, M.; Güvener, N.; Engelmann, U.; et al. Size-isolation of superparamagnetic iron oxide nanoparticles improves MRI, MPI and hyperthermia performance. Journal of Nanobiotechnology 2020, 18, 1-13, doi:10.1186/s12951-020-0580-1.

  1. Could the authors provide a histological study of the organs and tissues from the transplanted mice to confirm the effective transplantation? This will help to further validate the effectiveness of the transplantation and the bioluminescence imaging technique used in this study.

Answer:  Thank you for suggestion, our study sought to combine ICP-MS and bioluminescence imaging as a new tool to assess HSC migration and engraftment, indeed a histological study could be a complementary technique to validate transplant success, however, in the present study the validation was performed by blood count, which allowed monitoring the animal recovery in a non-invasive way and without the need of tissue collection, meanwhile histology would require the collection of the organs and would only present data on tissue cellularity.

Round 2

Reviewer 1 Report

The authors addressed all the raised concerns and significantly improved the quality of this manuscript. 

Reviewer 2 Report

The authors have addressed all my comments. Therefore, I recommend acceptance of this manuscript without further revisions.